# Improved health care utilization and costs in transplanted versus non-transplanted adults with sickle cell disease

**Santosh L. Saraf**[1]*, **Krishna Ghimire**[1], **Pritesh Patel**[1], **Karen Sweiss**[1,2], **Michel Gowhari**[1], **Robert E. Molokie**[1,3], **Victor R. Gordeuk**[1], **Damiano Rondelli**[1]*

**1** Division of Hematology & Oncology, Department of Medicine, University of Illinois at Chicago, Chicago, Illinois, United States of America, **2** Department of Pharmacy Practice, College of Pharmacy, University of Illinois at Chicago, Chicago, Illinois, United States of America, **3** Jesse Brown VA Medical Center, Chicago, Illinois, United States of America

* ssaraf@uic.edu (SLS); drond@uic.edu (DR)

**Data Availability Statement:** All relevant data are within the manuscript.

**Funding:** The authors received no specific funding for this work.

## Abstract

Patients with sickle cell disease (SCD) have access to fewer health care resources and therapies compared to other diseases, which contributes to increased morbidity and health care utilization. We compared health care utilization (inpatient hospital days, emergency care visits) and health care-related costs between SCD adults that underwent hematopoietic stem cell transplantation (HSCT) using a nonmyeloblative conditioning regimen versus those referred for HSCT but did not proceed due to lack of an HLA-matched sibling donor, denial by insurance, red blood cell antibodies to the potential donor, or declining further evaluation. Between 8/2011 and 4/2016, 83 SCD patients were referred for allogeneic HSCT and 16 underwent the procedure. The HSCT and non-HSCT groups were similar by age, sex, prior SCD-related therapy and complications. Compared to pre HSCT, significantly fewer inpatient hospital days (median of 1 versus 22 days, P = 0.003) and emergency care visits (median of 1 versus 4 visits, P = 0.04) were observed by the 2nd year post-HSCT. Similar results were observed in comparison to the standard-of-care group (median of 1 versus 12 hospital days, P = 0.002; median of 1 versus 3 emergency visits, P = 0.03). Lower health care costs were observed by the 2nd year post-HSCT (median of $16,281 versus $64,634 pre-HSCT (P = 0.01) and versus $54,082 in the standard-of-care group (P = 0.05). A median reduction of -$20,833/patient/year (IQR, -$67,078-+$4,442/patient/year) in health care costs compared to pre-HSCT was observed in the 2nd year post-HSCT. In conclusion, allogeneic HSCT leads to improvements in health care utilization and costs compared to standard-of-care therapy in high-risk SCD adults.

## Introduction

Sickle cell disease (SCD) is an inherited red blood cell disorder that affects approximately 1 in 365 African Americans at birth [1] and 25 million people worldwide.[2] Patients with SCD experience inferior health outcomes and have access to fewer health resources compared to

**Competing interests:** The authors have declared
that no competing interests exist.

other diseases.[3] A large majority of SCD patients are enrolled in Medicaid, which is accepted
by less than 70% of physicians in the U.S.[3] Suboptimal coverage for disease-appropriate
comprehensive care leads to patients with SCD relying on emergency room and inpatient hos-
pitalization settings for their medical care.[3, 4]

In parallel with the lack of access to comprehensive care, there is a substantial economic
burden for treating SCD patients in the emergency room and inpatient hospital settings. It is
currently estimated that $1.6 billion per year is spent in the United States of America on
healthcare related-costs for SCD-related complications.[5] Approximately 81% of these costs
are attributed to inpatient hospital care. Both health care related costs and health care utiliza-
tion (HCU) increase with older age in SCD.[5, 6] New therapies under development may
reduce the health disparities that SCD patients face and decrease patient morbidity and the
financial burden to the health care system.

Until recently, hydroxyurea was the only FDA-approved therapy available to treat patients
with SCD. Hydroxyurea reduces the rates of vaso-occlusive crises (VOC), acute chest syn-
drome, and red blood cell transfusion requirements [7] but the effects of hydroxyurea therapy
on health care costs have been mixed.[8–11] Voxelotor is a recently FDA-approved therapy
that improves hemoglobin concentration in patients with SCD, although its effects on SCD-
related complications and HCU are unclear.[12] Allogeneic hematopoietic stem cell transplan-
tation (HSCT) is a curative therapy for SCD that has been predominantly applied in children
due to concerns for higher rates of graft-versus host disease (GVHD) and lower rates of event-
free and overall survival in adults.[13] In children, improvements in inpatient lengths of stay
and health care costs are observed post-HSCT compared to pre-HSCT,[14, 15] although the
cost benefit of HSCT compared to standard of care is not clear.[14] Furthermore, increased
disease severity, which is in part defined by older age, and developing GVHD, are associated
with poorer outcomes, increased costs, and HCU post-HSCT.[14, 15] The effects of HSCT on
HCU and costs in SCD adults, who have greater SCD-related complications and a higher bur-
den of care on the health care system than children,[5] have not been reported.

Recent advances using non-myeloablative conditioning regimens with an HLA-matched
related donor have demonstrated that HSCT can be safely performed in SCD adults without a
high rate of severe complications and with the achievement of an event-free survival > 87%
and overall survival > 97%.[16, 17] Here we conducted a 2-year longitudinal analysis of adult
SCD patients with similar characteristics, stratified by those who received a NMA HSCT ver-
sus those who did not, to assess whether HSCT improved the financial burden of care in SCD
adults by comparing HCU and health care costs.

## Methods

The protocol was approved by the University of Illinois at Chicago (UIC) Institutional Review
Board prior to conducting the research. We analyzed SCD patients receiving their routine
medical care at the UIC Sickle Cell Center who were referred to the Blood & Marrow Trans-
plant Clinic between August 2011 and April 2016. One hundred and twenty-six SCD patients
meeting standard HSCT-eligibility criteria [18, 19] were referred during this time period, of
which 83 were internal referrals and received their routine care at the UIC Sickle Cell Center.
Patients who underwent a matched sibling donor HSCT received a non-myeloablative condi-
tioning regimen with alemtuzumab and single dose total body irradiation 3 Gy as previously
described.[17] Patients that did not proceed with HSCT continued on standard of care therapy
according to their primary SCD physician's discretion.

In SCD patients undergoing HSCT, clinical data was collected one year before the HSCT
date (pre-HSCT) and one and two years post-HSCT. In SCD patients that did not undergo

HSCT, clinical data was collected one and two years after the transplant consultation date. Data collected included: rates of vaso-occlusive crises requiring medical attention, acute chest syndromes, strokes, and red blood transfusions. Emergency room or acute care center visits and inpatient hospital days were quantified during the respective time periods. Healthcare costs were extracted using Compass to gather all patient activity and Trendstar to gather the cost data. The cost data included costs for inpatient and outpatient fees, medications, and diagnostic and laboratory testing. Physician fees were not included in the Trendstar data. Inpatient hospital days and the costs of the HSCT procedure were included in the 1-year post-HSCT category.

Clinical data, HCU, and costs in the HSCT group were compared pre-HSCT to the 1st and 2nd year post-HSCT using a Wilcoxon matched-pairs signed rank test or paired t-test analysis for linear variables and the Chi square test for categorical variables. Comparisons between the HSCT group and the non-HSCT group were performed by the Mann Whitney test and Chi square test for linear and categorical variables, respectively. A P-value $< 0.05$ was considered statistically significant. To determine clinical significance, odds ratios (OR) and β-values with 95% confidence intervals (CI) were calculated using logistic and linear regression analyses, respectively. Costs were log transformed for the linear regression analyses. Systat 11 (Systat Software Corporation, Chicago, IL, USA). Median and interquartile ranges (IQR) are provided.

## Results

Between August 2011 and April 2016, 83 SCD patients that received their routine medical care at our center were referred for HSCT. Of these, 16 proceeded with allogeneic HSCT while 67 did not. The reasons for not undergoing HSCT included lack of an HLA-matched related donor (n = 34, 51%), patient or family declining further work up for HSCT (n = 20, 30%), insurance denial (n = 11, 16%), or the presence of red blood cell antibodies to the potential donor (n = 2, 3%). The HSCT and non-HSCT SCD patients were similar with respect to age, sex, hemoglobin genotype, prior SCD-related therapy, and disease-related HSCT eligibility criteria (Table 1). Of the 16 SCD patients that proceeded to HSCT, 13 (81%) had stable donor engraftment and no patients developed acute or chronic GVHD. The median inpatient cost attributed to the HSCT was $92,666 (IQR, $71,735–$136,631) and the median length of hospitalization during the HSCT was 33 days (IQR, 24–30 days). The health care-related costs were

**Table 1. Baseline characteristics of patients with sickle cell disease that underwent hematopoietic stem cell transplantation (HSCT) vs. continued standard care.**

| Clinical Variable | Non-HSCT(n = 67) | HSCT(n = 16) | P Value |
|---|---|---|---|
| Age (years) | 34 (23–44) | 33 (24–34) | 0.2 |
| Male (%): Female (%) | 34%: 66% | 56%: 44% | 0.1 |
| Hb SS Genotype | 54 (81%) | 15 (94%) | 0.4 |
| Insurance Type | Medicaid: 37 (55%)<br>Medicare: 19 (28%)<br>Private: 11 (16%) | Medicaid: 5 (31%)<br>Medicare: 8 (50%)<br>Private: 3 (19%) | 0.2 |
| Sickle cell disease-related therapy | Hydroxyurea: 40 (60%)<br>Chronic RBC transfusions: 15 (22%) | Hydroxyurea: 10 (63%)<br>Chronic RBC transfusions: 5 (31%) | 0.8<br>0.5 |
| HSCT eligibility criteria | $\geq$ 3 VOC/year: 61 (91%)<br>$\geq$ 2 ACS/lifetime: 52 (78%)<br>Stroke: 17 (25%) | $\geq$ 3 VOC/year: 14 (88%)<br>$\geq$ 2 ACS/lifetime: 12 (75%)<br>Stroke: 4 (25%) | 0.7<br>0.8<br>1.0 |

RBC, red blood cell; VOC, vaso-occlusive crisis; ACS, acute chest syndrome

Median (interquartile range) provided

greater in the first year post-HSCT ($135,568, IQR $114,840–$205,853) compared to the year pre-HSCT ($64,634, IQR $24,354–$102,588).

## Transplant eligibility criteria are associated with increased costs in SCD

Among all 83 HSCT and non-HSCT patients, the median cost of care in the year prior to HSCT or consultation was $44,533 (IQR, $16,151–$126,473). Greater costs were observed in females versus males ($74,135 vs. $20,802; P = 0.004) (β 0.85, 95% CI: 0.15 to 1.54) and in those patients requiring chronic red blood cell transfusions ($113,437 vs. $31,120; P = 0.003) (β 1.08, 95% CI: 0.31 to 1.85). Statistically significant differences were not observed in costs by age (P = 0.2) (β 0.02, 95% CI: -0.01 to 0.05) (Fig 1A), hemoglobin genotype (HbSS: $50,606 vs. non-HbSS: $27,626; P = 0.2) (β 0.65, 95%CI: -0.25 to 1.57), hydroxyurea therapy (HU: $49,103 vs. no HU: $41,396; P = 0.8) (β -0.15, 95% CI: -0.92 to 0.62), or insurance type (Medicaid: $42,315 vs. Medicare: $67,689 vs. Private insurance: $29,581; P = 0.3) (Private insurance vs. Medicaid or Medicare β -0.43, 95% CI: -1.34 to 0.48). Health care costs were directly associated with the number of inpatient days (β 0.03, 95% CI: 0.02 to 0.04) (Fig 1B). In addition, higher costs were observed in those SCD patients with ≥ 3 VOC versus < 3 VOC (β 1.77, 95% CI:

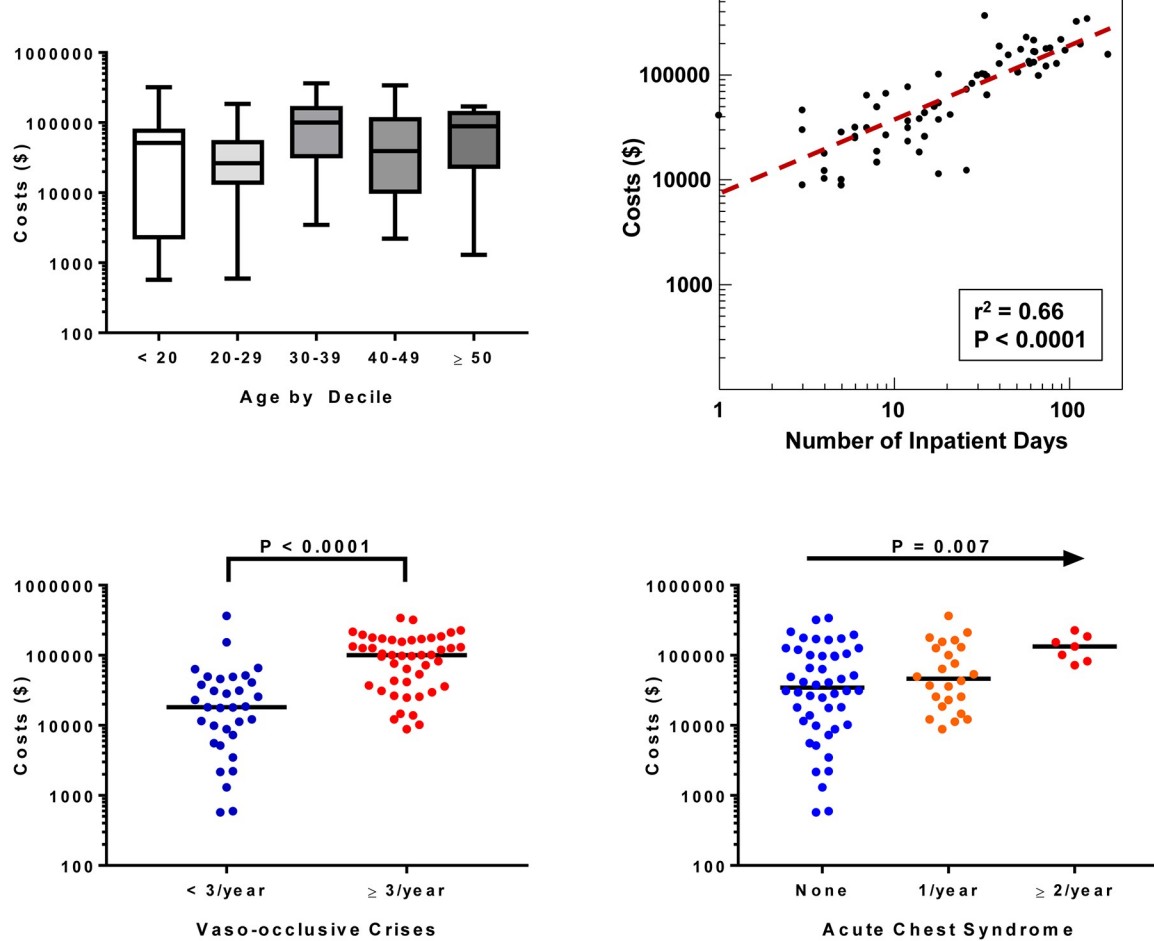

**Fig 1. Clinical variables associated with health care costs in 83 patients with sickle cell disease in the year prior to evaluation for HSCT.**

**Table 2. Sickle cell disease-related complications after hematopoietic stem cell transplantation (n = 16).**

| Complication | Pre-HSCT 1 Year | Post-HSCT 1st Year | Post-HSCT 2nd Year | P Value (Year 1 vs. pre-HSCT) | P Value (Year 2 vs. pre-HSCT) |
|---|---|---|---|---|---|
| RBC Transfusion | 2 (0–26) | 8 (6–16) | 0 (0–13) | 0.4 | 0.005 |
| Vaso-occlusive crisis | 4 (1–29) | 2 (0–8) | 0 (0–37) | 0.1 | 0.02 |
| Acute chest syndrome | 5 (31%) | 1 (6%) | 0 (0%) | 0.07 | 0.02 |
| Stroke | 2 (13%) | 0 (0%) | 1 (6%) | 0.1 | 0.6 |

13/16 had stable engraftment; 3/16 had secondary graft loss

HSCT, hematopoietic stem cell transplantation; RBC, red blood cell

Median (range) value provided

1.19 to 2.35) (Fig 1C), acute chest syndrome versus no acute chest syndrome (β 0.66, 95% CI: 0.2 to 1.11) (Fig 1D), and stroke versus no stroke (stroke: $134,112 vs. no stroke: $41,144; P = 0.05) (β 1.43, 95% CI: -0.14 to 2.99) during that year.

## Reduced HCU and costs in the transplanted cohort second year post-transplant compared to year pre-transplant

Compared to the year pre-HSCT, rates of red blood cell transfusions (2 units, IQR 0–26 units vs. 0 units, IQR 0–13 units, respectively; P = 0.005) (OR for transfusion 0.39, 95% CI: 0.09 to 1.78), vaso-occlusive crises (4 episodes, IQR 1–29 episodes vs. 0 episodes, IQR 0–37 episodes, respectively; P = 0.02) (OR for vaso-occlusive crisis 0.13, 95% CI: 0.02 to 0.75), and acute chest syndromes (31% of patients affected vs. 0% of patients affected, respectively; P = 0.02) (OR not calculatable), were lower by the 2nd year post-HSCT (Table 2). Consistent with the improvements in SCD-related complications, emergency room visits (OR for emergency room visit 0.07, 95% CI: 0.01 to 0.63) and inpatient hospital days (OR for hospitalization 0.14, 95% CI: 0.02 to 0.84) were also lower by the 2nd year post-HSCT (Fig 2).

By the second year post-transplant, the health care-related costs were lower compared to the year pre-HSCT (pre-HSCT: $64,634, IQR $24,354 - $102,588; 2nd year post-HSCT: $16,281, IQR $5,471 to $58,298; P = 0.01) (β -1.01, 95% CI: -1.95 to -0.05) with a median

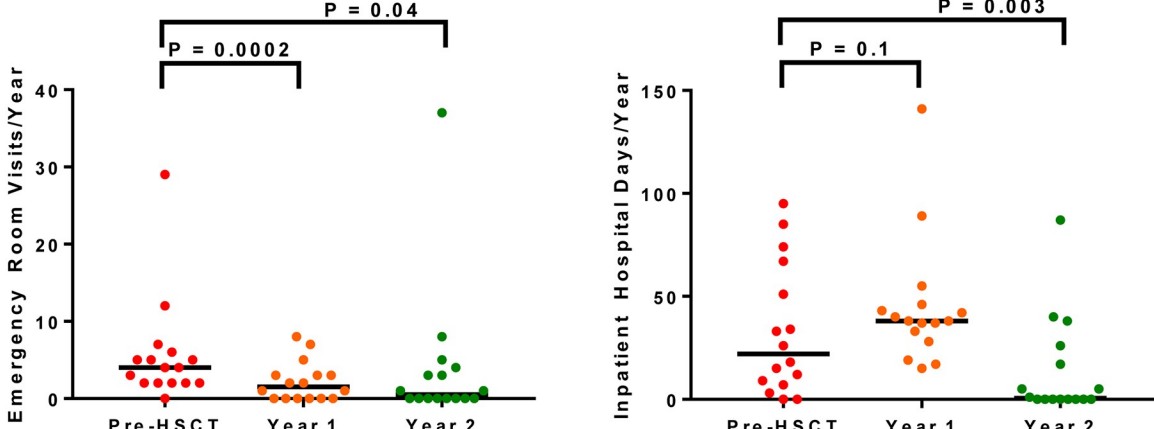

**Fig 2. Health care utilization.** (A) Emergency visits improved from pre-HSCT (4 visits, IQR 2–6 visits) to 1 year (2 visits, IQR 0–3 visits) and 2 years post-HSCT (1 visit, IQR 0–4 visits). (B) The number of inpatients days increased from pre-HSCT (22 days, IQR 8–59 days) to 1 year post-HSCT (38 days, IQR 31–45 days) and then improved by the 2nd year post-HSCT (1 day, IQR 0–22 days).

reduction of $20,833/patient/year (IQR, -$67,078–+$4,442/patient/year). Health care related-costs were significantly lower in the 13 patients that had stable engraftment compared to the 3 patients with secondary graft loss at 1-year ($123,796 vs. $283,596, respectively; P = 0.02) (β -0.84, 95% CI: -1.55 to -0.13) and 2-years post-HSCT ($7,471 vs. $374,591, respectively; P = 0.004) (β -2.74, 95% CI: -4.44 to -1.05).

### Reduced HCU and costs in the 2nd year post-transplant compared to standard of care

In the first year, we observed lower emergency room visits and increased inpatient hospital days in the HSCT versus non-HSCT group (Fig 3). By the second year, both emergency room visits (OR for emergency room visit 0.09, 95% CI: 0.02 to 0.34) and inpatient hospital days (OR for hospitalization 0.17, 95% CI: 0.05 to 0.55) were lower in the HSCT versus the non-HSCT group (Fig 3). Health care costs remained relatively stable in the year before, the first year after, and the second year after consultation in the non-HSCT group (Fig 4). Consistent with the HCU data, health care-related costs were higher in the first year post-transplant (HSCT: $135,568, IQR $114,840–$205,583; non-HSCT: $47,437, IQR $15,264–$140,375; P = 0.0004) (β 1.31, 95% CI: 0.66 to 1.97) but lower in the second year post-transplant (HSCT: $16,281, IQR $5,471–$58,298; non-HSCT: $54,082, IQR $18,294–$126,748; P = 0.05) (β -0.62, 95% CI: -1.51 to 0.28) compared to the non-HSCT group (Fig 4).

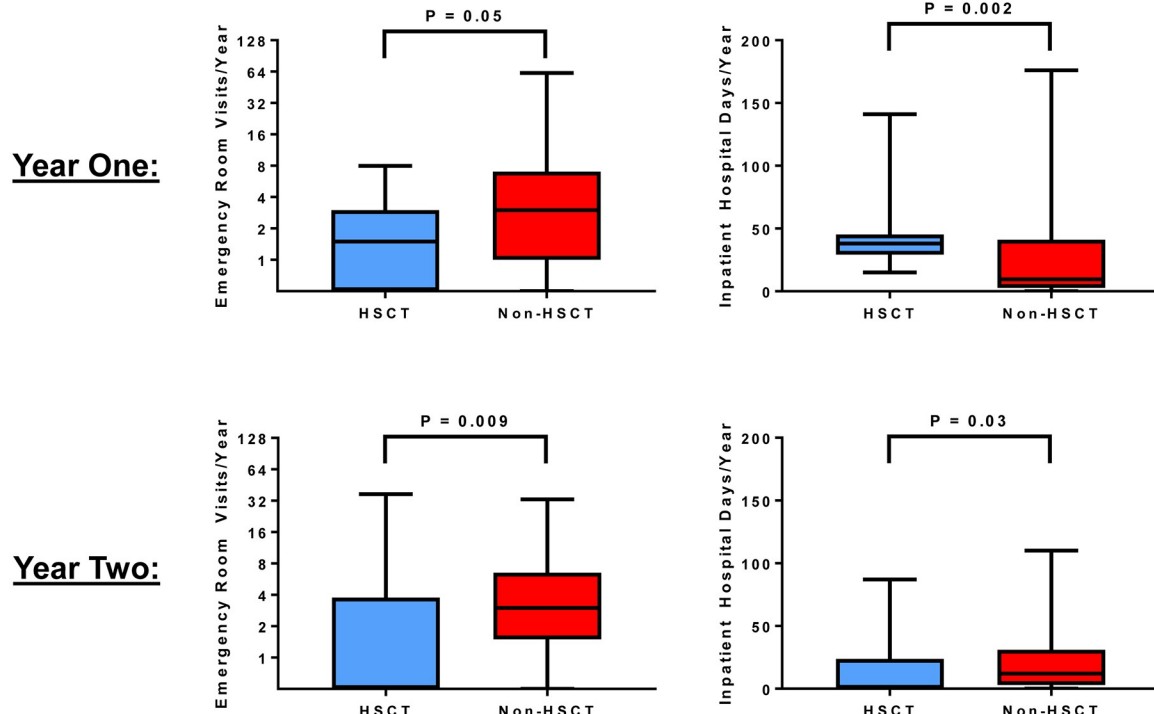

**Fig 3. Comparison of emergency care and inpatient hospital days in 16 transplanted and 67 non-transplanted patients with sickle cell disease.** (A) At 1 year from transplantation or the time of referral in the non-transplanted patients, emergency visits were lower (2 visits, IQR 0–3 visits vs. 3 visits, IQR 1–7 visits, respectively) while inpatient hospital days were higher (38 days, IQR 30–45 days vs. 10 days, IQR 3–39 days, respectively) in the HSCT vs. non-HSCT groups. (B) By the 2nd year, improvements in both emergency room visits (1 visit, IQR 0–4 visits vs. 3 visits, IQR 2–7 visits, respectively) and inpatients hospital days (1 day, IQR 0–22 days vs. 12 days, IQR 3–31 days, respectively) were observed in the HSCT vs. non-HSCT groups.

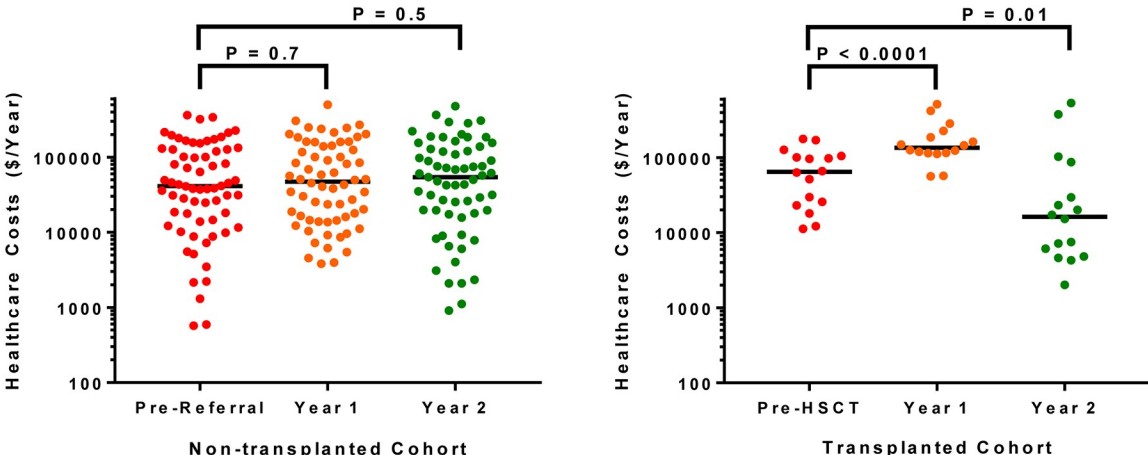

**Fig 4. Health care costs. (A)** Health care costs remained relatively stable in the non-HSCT group from pre-referral ($41,144, IQR $14,590–$126,508), 1 year post-referral ($47,437, IQR $15,264–$140,375), and 2 years post-referral ($54,082, IQR $18,294–$126,748). **(B)** Health care costs increased from pre-HSCT values ($64,634, IQR $24,354–$102,588) to the 1st year post-HSCT ($135,568, IQR $114,840–$205,853) and then improved by the 2nd year post-HSCT ($16,281, IQR $5,471–$58,298).

## Discussion

Sickle cell disease causes acute and chronic complications that lead to substantial morbidity. The clinical course of SCD is further complicated by a lack of access to health care resources and treatment options. In a longitudinal cohort of adult SCD patients, we demonstrate that allogeneic HSCT using a nonmyeloablative conditioning approach leads to reduced SCD-related complications, HCU and costs by the second year post-HSCT compared to pre-HSCT values. Furthermore, we demonstrate that HCU and costs are lower in those SCD patients that proceeded to HSCT compared to SCD patients with similar SCD-related complications that did not undergo HSCT and continued on standard of care therapy.

It is estimated that the average cost of care for an adult with SCD ranges from $21,720 to $58,044 per year.[5, 20] Approximately 84% of these costs are attributed to SCD-related complications and 81% are due to costs incurred during hospitalizations.[5] Consistent predictors for increased HCU and costs in the literature have included older age, female gender, acute chest syndrome, stroke, red blood cell transfusions, public insurance, and the number of hospitalizations.[20–26] We found that the median cost of care for UIC adults with SCD in the year prior to referral for HSCT was $44,533 with an interquartile range of $16,151 –$126,473. Higher costs of care were associated with red blood cell transfusion requirements and inpatient hospital days. Furthermore, common definitions for clinically aggressive SCD that warrant disease modifying therapy and are used as indicators for HSCT evaluation,[18, 19] such as ≥ 3 VOC/year, acute chest syndrome, and stroke, were associated with higher costs. Understanding the estimates and predictors for HCU and costs in high risk SCD adults may help guide cost analyses for emerging interventions, such as selectin inhibitors[27, 28] and gene therapy. [29]

New in our study is the finding that allogeneic HSCT using a nonmyeloablative conditioning approach with an HLA-matched sibling donor improves HCU and costs in adults with SCD as soon as two years after transplant. A recent outcome analysis of allogeneic HSCT in children with SCD showed 5-year event-free and GVHD-free survival rates of 93% and 86%, respectively.[13] This has led to improvements in inpatient hospital days[14] and costs[15] in SCD children pre-HSCT compared to after HSCT. However, no significant differences in

health care related costs were observed between those SCD children that underwent HSCT compared to SCD children that did not undergo HSCT.[14] This differs from our longitudinal study in adult SCD patients and may be due to the use of HLA-matched siblings versus unrelated donors, using peripherally mobilized rather than cord blood stem cells, and the absence of acute and chronic GVHD with our nonmyeloablative HSCT approach.

In our cohort of SCD adults that were referred to our transplant clinic and received a transplant, we observed improvements in acute care utilization, inpatient hospital days, and health care costs by the second year post-HSCT compared to pre-HSCT levels. We also compared HCU and costs between the cohort of patients who underwent HSCT versus the cohort of SCD adults that were referred to transplant but did not proceed and received standard of care therapy. In these two cohorts, that were comparable with regards to age, insurance type, prior therapy, and SCD severity, we demonstrated that SCD adults undergoing HSCT had lower acute care utilization, inpatient hospital days, and health care costs by the second year post-transplant compared to those that continued with standard of care therapy. To our knowledge, this is the first reported analysis for the economic impact of HSCT in adults with SCD. Based on these results and on a reasonable expectation for an incremental cost benefit over a prolonged observation time, we believe that our results may help guide the decision process for policy makers and insurance providers. It is estimated that patients with SCD have $695,000 of lost income over their lifetime due to SCD-related complications and early mortality.[30] Hematopoietic stem cell transplantation may help regain this lost income of individuals with SCD, on top of the suggested improvement in HCU, and this will need to be investigated in future studies.

Limitations to our study include that HCU and costs that were incurred outside of our institution were not included, although SCD patients that underwent HSCT almost exclusively received their post-HSCT care at our institution. Subgroup analyses based on age and sex will need to be conducted in larger cohorts. Future investigation in a multicenter study is warranted to investigate the long-term effects of HSCT on HCU and costs in adults with SCD.

In conclusion, allogeneic HSCT may lead to improvements in HCU and costs compared to standard-of-care therapy in high-risk SCD adults. With a nonmyeloablative HSCT approach, the costs of the HSCT can be offset by the reductions in HCU and costs after approximately six years and may lead to improvements in both the morbidity and the financial burden on the health care system in this high-risk SCD patient group. Patients with SCD have limited access to health resources and therapeutic interventions, leading to poor health outcomes. Our findings highlight a therapeutic intervention, allogeneic HSCT, which leads to improved health and reduced health care utilization in the emergency room and inpatient settings for patients with SCD. Developing strategies to overcome barriers to allogeneic HSCT may help improve the health equity in patients with SCD.

## Acknowledgments

We thank the individuals for their participation in this study as well as the staff and transplant coordinators for their efforts.

## Author Contributions

**Conceptualization:** Santosh L. Saraf, Pritesh Patel, Karen Sweiss, Michel Gowhari, Robert E. Molokie, Victor R. Gordeuk, Damiano Rondelli.

**Data curation:** Santosh L. Saraf, Krishna Ghimire.

**Formal analysis:** Santosh L. Saraf, Krishna Ghimire, Pritesh Patel, Damiano Rondelli.

**Investigation:** Santosh L. Saraf, Krishna Ghimire, Karen Sweiss, Michel Gowhari, Robert E. Molokie, Victor R. Gordeuk, Damiano Rondelli.

**Methodology:** Santosh L. Saraf, Krishna Ghimire, Pritesh Patel, Karen Sweiss, Michel Gowhari, Robert E. Molokie, Victor R. Gordeuk, Damiano Rondelli.

**Supervision:** Santosh L. Saraf.

**Writing – original draft:** Santosh L. Saraf, Krishna Ghimire, Pritesh Patel, Karen Sweiss, Michel Gowhari, Robert E. Molokie, Victor R. Gordeuk, Damiano Rondelli.

**Writing – review & editing:** Santosh L. Saraf, Krishna Ghimire, Pritesh Patel, Karen Sweiss, Michel Gowhari, Robert E. Molokie, Victor R. Gordeuk, Damiano Rondelli.

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
