## [Decision Letter · Decision Letter 0]

27 Dec 2019

PONE-D-19-30828

Improved Health Care Utilization and Costs in Transplanted versus Non-transplanted Adults with Sickle Cell Disease

PLOS ONE

Dear Dr.Saraf,

Thank you for submitting your manuscript to PLOS ONE. After careful consideration, we feel that it has merit but does not fully meet PLOS ONE’s publication criteria as it currently stands. Therefore, we invite you to submit a revised version of the manuscript that addresses the points raised during the review process.

We have now received reports from two referees of your manuscript, as agree with reviewers comments raised a few concerns about this study. After careful consideration, we invite you to submit a revised version of the manuscript.  

We would appreciate receiving your revised manuscript by Feb 10 2020 11:59PM. To enhance the reproducibility of your results, we recommend that if applicable you deposit your laboratory protocols in protocols.io, where a protocol can be assigned its own identifier (DOI) such that it can be cited independently in the future. For instructions see: http://journals.plos.org/plosone/s/submission-guidelines#loc-laboratory-protocols

We look forward to receiving your revised manuscript.

Kind regards,

Senthilnathan Palaniyandi, Ph.D

Academic Editor

PLOS ONE

Journal Requirements:

Reviewers' comments:

Reviewer's Responses to Questions

**Comments to the Author**

1. Is the manuscript technically sound, and do the data support the conclusions?

Reviewer #1: Yes

Reviewer #2: Yes

2. Has the statistical analysis been performed appropriately and rigorously? 

Reviewer #1: Yes

Reviewer #2: No

3. Have the authors made all data underlying the findings in their manuscript fully available?

Reviewer #1: Yes

Reviewer #2: Yes

4. Is the manuscript presented in an intelligible fashion and written in standard English?

Reviewer #1: Yes

Reviewer #2: Yes

5. Review Comments to the Author

Reviewer #1: Thank you for the opportunity to review this work from Dr. Saraf and his colleagues. The authors in this manuscript are trying to provide an evidence on the improvement of health care utilization (HCU) and the associated costs in patients who were diagnosed with sickle cell disease (SCD) and underwent stem cell transplant compared to patients who received standard of care. Overall, the used methods are appropriate and the manuscript is well written. The following are my comments:

1- For the conclusion, I recommend to adjust the language the conclusion to be as following: “In conclusion, allogeneic HSCT may leads to improvements in health care utilization and costs compared to standard-of-care therapy in high risk SCD adults.” Given the relative small size and the need of multicenter study to replicate these findings before providing a such strong recommendation.

2- In abstract, “HSCT” needs to be defined for one time in the beginning.

3- This article published in JAMA recently which underscores the need for disease-modifying therapies to improve the underlying morbidity and mortality associated with SCD. You might use it to justify the importance of transplant as a disease modifying therapy and how could improve the lifetime income of individuals with SCD on top of the suggested improvement in HCU in the current presented evidence (economic benefits on multiple levels). https://jamanetwork.com/journals/jamanetworkopen/fullarticle/2755485?utm_source=silverchair&utm_medium=email&utm_campaign=article_alert-jamanetworkopen&utm_content=mthlyforyou&utm_term=120819&term=120819

4- In the introduction, hydroxyurea is mentioned as the only FDA approved therapy available to treat patients with SCD. Now, since voxelotor has been recently approved by FDA in November 22, 2019 (which I understand that happened after submitting this manuscript), I highly recommend to comment on this medication and the related benefits as another option for patients with SCD.

5- I recommend to define more in the inclusion criteria the exact meaning of “match related HSCT”. Does that mean matched sibling or matched parents or other related family members or combination of all of them? And I recommend to report the percentage of each category in case we have a combination.

6- It is unclear the age range that included in this study. The authors reported median and interquartile range (IQR). The IQR for HSCT group is 16-51. I assume this means there are some children who aged less than 16 and included in this study. I recommend to clarify this point to know what kind of population we are dealing with in this study and to report the percentages of children (less than or equals 16) and adults (more than 16).

7- The current literature suggests that myeloablative conditioning has good outcomes on survival, GVHD and engraftment with even possible superiority of myeloablative over nonmyeloablative/reduced intensity conditioning (RIC) in children population and subsequently possible less burden on health care system, whereas nonmyeloablative has better outcomes in adults. I suggest to put a rationale for limiting the data on RIC since we are talking about possible combined population in this evidence.

8- Proceeded not proceed in page 11, line 8.

9- In the legend of figure 1, I suggest to clarify the population and the time frame you are trying to investigate. I believe these outcomes reported on all patients in the year prior to HSCT based on the results section.

Reviewer #2: 1. The authors have done a fair job in presenting the data. Authors have described the statistical analyses used on the data, but did not indicate the level of significance when comparing different groups on the data. Although the P values are mentioned absence of indication of level of significance leads to undermining and confusion of the data presented. In a study that focuses on comparison of clinical data over a period of time, analyses and representation needs to be precise and clearly stated.

2. Clinical level of significance should be mentioned in all the data that is presented.

3. Is there a reason why there were greater costs observed in female group of patients versus male group of patients? Did the authors compare HSCT Vs non HSCT patients based on gender?

6. PLOS authors have the option to publish the peer review history of their article (what does this mean?). If published, this will include your full peer review and any attached files.

Reviewer #1: Yes: Jehad Almasri

Reviewer #2: No

---

## [Author Response · Author response to Decision Letter 0]

9 Jan 2020

Reviewer #1: 

1- For the conclusion, I recommend to adjust the language the conclusion to be as following: “In conclusion, allogeneic HSCT may leads to improvements in health care utilization and costs compared to standard-of-care therapy in high risk SCD adults.” Given the relative small size and the need of multicenter study to replicate these findings before providing a such strong recommendation.

Thank you for this recommendation. As suggested, we have changed our language in the conclusion (page 13, 3rd paragraph, 1st sentence).

2- In abstract, “HSCT” needs to be defined for one time in the beginning.

We have provided the definition for HSCT in the abstract (page 3, 2nd sentence).

3- This article published in JAMA recently which underscores the need for disease-modifying therapies to improve the underlying morbidity and mortality associated with SCD. You might use it to justify the importance of transplant as a disease modifying therapy and how could improve the lifetime income of individuals with SCD on top of the suggested improvement in HCU in the current presented evidence (economic benefits on multiple levels). https://hes32-ctp.trendmicro.com:443/wis/clicktime/v1/query?url=https%3a%2f%2fjamanetwork.com%2fjournals%2fjamanetworkopen%2ffullarticle%2f2755485%3futm%5fsource%3dsilverchair%26utm%5fmedium%3demail%26utm%5fcampaign%3darticle%5falert%2djamanetworkopen%26utm%5fcontent%3dmthlyforyou%26utm%5fterm%3d120819%26term%3d120819&umid=3f7b50fa-59c4-41b9-913d-a3736bdaf5f3&auth=85c5a955287d1e42fab58bed777dfa626e5ad059-916f504479157f4df090472dc4676f843ea59603

We appreciate the reviewer providing this important reference, which we have now included in the discussion as an additional potential benefit of transplantation in patients with SCD (page 13, 1st paragraph, 1st and 2nd sentences).

4- In the introduction, hydroxyurea is mentioned as the only FDA approved therapy available to treat patients with SCD. Now, since voxelotor has been recently approved by FDA in November 22, 2019 (which I understand that happened after submitting this manuscript), I highly recommend to comment on this medication and the related benefits as another option for patients with SCD.

In agreement, we have included voxelotor as a recently FDA-approved therapy to the introduction (page 4, 3rd paragraph, 3rd sentence).

5- I recommend to define more in the inclusion criteria the exact meaning of “match related HSCT”. Does that mean matched sibling or matched parents or other related family members or combination of all of them? And I recommend to report the percentage of each category in case we have a combination.

To clarify the inclusion, we have changed the terminology to matched sibling donor in the methods section (page 6, 1st paragraph, 4th sentence). All of the donors were matched sibling donors.

6- It is unclear the age range that included in this study. The authors reported median and interquartile range (IQR). The IQR for HSCT group is 16-51. I assume this means there are some children who aged less than 16 and included in this study. I recommend to clarify this point to know what kind of population we are dealing with in this study and to report the percentages of children (less than or equals 16) and adults (more than 16).

Thank you for bringing up this error – we had accidentally provided the range (16 – 51 years) in the prior draft and have corrected the table to provide the interquartile range (24 – 34 years) in the revised draft (page 19, Table 1). All of the patients in this analysis are age 16 or older.

7- The current literature suggests that myeloablative conditioning has good outcomes on survival, GVHD and engraftment with even possible superiority of myeloablative over nonmyeloablative/reduced intensity conditioning (RIC) in children population and subsequently possible less burden on health care system, whereas nonmyeloablative has better outcomes in adults. I suggest to put a rationale for limiting the data on RIC since we are talking about possible combined population in this evidence.

Because this analysis is predominantly an adult cohort (all age 16 or older), we have focused on those conditioning regimens that have demonstrated safety and efficacy in adults with SCD and avoided discussion on myeloablative vs. nonmyeloablative/RIC regimens in children populations. As mentioned above in item #6, this was not a combined population of children and adults.

8- Proceeded not proceed in page 11, line 8.

We have changed “proceed” to “proceeded” (page 11, 1st paragraph, 4th sentence).

9- In the legend of figure 1, I suggest to clarify the population and the time frame you are trying to investigate. I believe these outcomes reported on all patients in the year prior to HSCT based on the results section.

Thank for bringing up this suggestion. We have clarified the population (all 83 patients) and the time frame (in the year prior to evaluation for transplantation) in the figure legend (page 18).

Reviewer #2: 

1. The authors have done a fair job in presenting the data. Authors have described the statistical analyses used on the data, but did not indicate the level of significance when comparing different groups on the data. Although the P values are mentioned absence of indication of level of significance leads to undermining and confusion of the data presented. In a study that focuses on comparison of clinical data over a period of time, analyses and representation needs to be precise and clearly stated.

Thank you for this comment and in the revised manuscript, we have provided the level of statistical significance for the analyses in the methods section (page 7, 1st paragraph, 3rd sentence).

2. Clinical level of significance should be mentioned in all the data that is presented.

As recommended, we have included measures of clinical significance (odds ratios for clinical outcomes, and beta values for cost as a linear variable). The methods for determining the clinical levels of significance are provided (page 7, 1st paragraph, 4th sentence) and the corresponding values have been provided in all of the relevant data in the results sections.

3. Is there a reason why there were greater costs observed in female group of patients versus male group of patients? Did the authors compare HSCT Vs non HSCT patients based on gender?

In this study, it is not clear why there were greater costs in females versus males. This may be due to either selection bias or a true biological difference. We have included in the discussion that this will need to be investigated further in larger, cohorts (page 13, 2nd paragraph, 2nd sentence).

---

## [Decision Letter · Decision Letter 1]

13 Feb 2020

Improved Health Care Utilization and Costs in Transplanted versus Non-transplanted Adults with Sickle Cell Disease

PONE-D-19-30828R1

Dear Dr. Saraf,

We are pleased to inform you that your manuscript has been judged scientifically suitable for publication and will be formally accepted for publication once it complies with all outstanding technical requirements.

With kind regards,

Senthilnathan Palaniyandi, Ph.D

Academic Editor

PLOS ONE

Additional Editor Comments (optional):

Reviewers' comments:

Reviewer's Responses to Questions

**Comments to the Author**

1. If the authors have adequately addressed your comments raised in a previous round of review and you feel that this manuscript is now acceptable for publication, you may indicate that here to bypass the “Comments to the Author” section, enter your conflict of interest statement in the “Confidential to Editor” section, and submit your "Accept" recommendation.

Reviewer #1: (No Response)

Reviewer #2: All comments have been addressed

2. Is the manuscript technically sound, and do the data support the conclusions?

Reviewer #1: Yes

Reviewer #2: Yes

3. Has the statistical analysis been performed appropriately and rigorously? 

Reviewer #1: Yes

Reviewer #2: Yes

4. Have the authors made all data underlying the findings in their manuscript fully available?

Reviewer #1: Yes

Reviewer #2: Yes

5. Is the manuscript presented in an intelligible fashion and written in standard English?

Reviewer #1: Yes

Reviewer #2: Yes

6. Review Comments to the Author

Reviewer #1: Two minor points:

1- The conclusion language need to be modified in the abstract as well.

2- I would still highly recommend to add the actual age range in the manuscript. I understand your range is 16-51 but I do not see this has been mentioned any where in the manuscript.

Reviewer #2: (No Response)

7. PLOS authors have the option to publish the peer review history of their article (what does this mean?). If published, this will include your full peer review and any attached files.

Reviewer #1: Yes: Jehad Almasri

Reviewer #2: No

---

## [Editor Report · Acceptance letter]

18 Feb 2020

PONE-D-19-30828R1 

Improved Health Care Utilization and Costs in Transplanted versus Non-transplanted Adults with Sickle Cell Disease 

Dear Dr. Saraf:

I am pleased to inform you that your manuscript has been deemed suitable for publication in PLOS ONE. Congratulations! Your manuscript is now with our production department. 

With kind regards,

on behalf of

Dr. Senthilnathan Palaniyandi 

Academic Editor

PLOS ONE